# Energy Storage Ceramics: A Bibliometric Review of Literature

**DOI:** 10.3390/ma14133605

**Published:** 2021-06-28

**Authors:** Haiyan Hu, Aiping Liu, Yuehua Wan, Yuan Jing

**Affiliations:** 1Library, Hangzhou Dianzi University, Hangzhou 310018, China; emmahhy@hdu.edu.cn; 2Center for Optoelectronics Materials and Devices, Zhejiang Sci-Tech University, Hangzhou 310018, China; liuaiping1979@gmail.com; 3Institute of Information Resource, Zhejiang University of Technology, Hangzhou 310014, China; wanyuehua@zjut.edu.cn; 4Library, Zhejiang Sci-Tech University, Hangzhou 310018, China

**Keywords:** energy storage ceramics, bibliometric, lead-free, microstructure, keywords analysis

## Abstract

Energy storage ceramics is among the most discussed topics in the field of energy research. A bibliometric analysis was carried out to evaluate energy storage ceramic publications between 2000 and 2020, based on the Web of Science (WOS) databases. This paper presents a detailed overview of energy storage ceramics research from aspects of document types, paper citations, h-indices, publish time, publications, institutions, countries/regions, research areas, highly cited papers, and keywords. A total of 3177 publications were identified after retrieval in WOS. The results show that China takes the leading position in this research field, followed by the USA and India. Xi An Jiao Tong Univ has the most publications, with the highest h-index. J.W. Zhai is the most productive author in energy storage ceramics research. *Ceramics International*, *Journal of Materials Science-Materials in Electronics*, and the *Journal of Alloys and Compounds* are the most productive journals in this field, and materials science—multidisciplinary is the most frequently used subject category. Keywords, highly cited papers, and the analysis of popular papers indicate that, in recent years, lead-free ceramics are prevalent, and researchers focus on fields such as the microstructure, thin films, and phase transition of ceramics.

## 1. Introduction

Energy storage ceramics are an important material of dielectric capacitors and are among the most discussed topics in the field of energy research [1]. Mainstream energy storage devices include batteries, dielectric capacitors, electrochemical capacitors, and fuel cells. Due to the low dielectric loss and excellent temperature, the status of ceramics is constantly highlighted. To our knowledge, the concept of energy storage ceramics has a long history. Some early papers on energy storage ceramics research were put forward in the mid-20th century. It is found that researchers worked on antiferroelectric ceramics with field-enforced transitions in 1961 [2], strontium titanate films in 1969 [3], glass-bonded lead zirconate in 1971 [4], and energy storage in ceramic dielectrics in 1972 [5]. Energy storage ceramics are considered to be a preferred material of energy storage, due to their medium breakdown field strength, low dielectric loss, antifatigue, and excellent temperature stability [6]. However, ceramic capacitors have not been considered for energy storage applications for a long time. The primary reasons for this are the expensive costs and the low energy density of large ceramic capacitors [7]. Researchers have been committed to improving the performance of energy storage ceramics and reducing their cost. Oily wastes and other residues have been applied in ceramic material manufacturing since 1988 [8]. Ferroelectric ceramics were introduced in composites to enhance their charge storage properties, and the dielectric and charge storage properties of these composites were studied in the 1990s [9]. The dielectric breakdown strength and other capabilities of ceramic material have been optimized over the years [10]. Researchers have also worked on the optimization of ceramic capacitors’ energy storage density, based on the Devonshire’s Theory of Ferroelectrics [11]. With the growth in energy demand, the potential applications of energy storage ceramics in the energy-storage area have been excavated. Currently, energy storage ceramics with higher energy densities and lower costs [12,13] are widely used in aerospace [14], military [15], oil drilling [16], and various applications.

Several reviews focus on energy storage ceramics. Researchers have analyzed the progress of sol–gel-derived composite ceramic carbon electrodes [17], ceramic membranes [18], conductivities of solid electrolyte materials in lithium-ion batteries [19], high-temperature sodium batteries [20], lead zirconate-based antiferroelectric materials [21], antiferroelectric ceramics capacitors [22,23], graphene-based materials for supercapacitor electrodes [24], solid-state electrolyte materials [25], lead-free dielectric ceramics [26,27], and high-strain perovskite piezoelectric ceramics [28].

Review papers can synthesize the key theories of a special topic of energy storage ceramics research. Different from review papers, bibliometric methods can analyze massive papers, and show the overall picture of energy storage ceramics research from the perspective of the literature. 

Bibliometrics was defined as the “statistical analysis of written publications, such as books or articles” by the OECD [29]. Bibliometric analysis is a statistical evaluation of published papers and academic research [30]. The development of modern bibliometric techniques can be traced back to 1896; Pareto published the first bibliometric paper [31]. More scholars, including Lotka [32], Zipf [33], Bradford [34], and Price [35], have developed new bibliometric methods since then. 

Bibliometric analysis provides a perspective that can easily be scaled from the micro- to macrolevel. It has been used to quantitatively analyze academic publications, to show the research status and trends in many research fields, such as health care science services [36,37,38,39,40], computer science [41,42], mechanical engineering [43,44,45,46], psychology [47,48], economics [49,50], energy [51,52], and ecology [53,54,55]. The United Kingdom has considered using bibliometrics in its research excellence framework, to assess the quality of research output [56]. 

To our knowledge, this work is not the first to assess the energy storage field using bibliometric methods, but is the first bibliometric analysis dedicated to research on energy storage ceramics. Previous bibliometric analysis has dealt with the international development trend of energy storage technology [57], research progress of lead-free dielectric ceramics, and emerging topics in energy storage [58], but the specialized and systematic study of energy storage ceramics research has not been reported to date. The aim of this research is to (1) provide an overview of this field; (2) find the leading countries/regions, institutes, and authors; (3) create an opportunity for cooperation between countries, institutions, and authors; (4) find the most productive journals; and (5) find popular topics, top papers, and research trends.

The remainder of this paper is arranged as follows: a section on the method and materials; a section on the results, analysis of the leading countries/regions, institutions, authors, publications, research areas, and keywords; a discussion of our findings; and a summary of this paper.

## 2. Materials and Methods

This analysis is based on the publications related to energy storage ceramics published between 2000 and 2020. Papers were collected from the Web of Science (WOS), with the search formula of “energy storage ceramic*” or “lead-free ceramic*” or “dielectric ceramic*”. Before the formal retrieval, we searched in Scopus, Google Scholar, Baidu Scholar, and WOS; the results showed that WOS can meet our needs. Furthermore, other terms, such as “ferroelectric material”, were used to expand the dataset and to ensure the result of retrieval was satisfactory. Our search was limited to the Web of Science (WOS) Core Collection, supported by the following three closely related databases: Science Citation Index-Expanded (SCI-E), Emerging Sources Citation Index (ESCI), and Conference Proceedings Citation Index-Science (CPCI-S). The search date was 11 January 2021, and the search field was restricted to “topic” (a paper’s title, abstract, author keywords, and keywords plus).

We used the Derwent Data Analyzer 10 (DDA10.0 build 27330, Search Technology Inc., Norcross, GA, USA) as the analytical tool. DDA is a data-mining platform that converts patent data, scientific literature, and business intelligence into actionable, commercial insight. The papers retrieved were organized into tables and DDA charts. Tables were produced to show the output, collaboration, and influence of countries/regions, institutions, and authors, as well as highly cited papers of this research field; a line chart was used to illustrate the publication trend of the research field and the top 3 most productive countries/regions; DDA cluster maps were employed to explain the collaborative relationships among countries/regions, institutions, and research areas; bubble charts were adopted to more intuitively show the development trends of journals and author keywords in energy storage ceramics research. The journal’s impact factor (IF) was determined according to the 2019 Journal Citation Reports (JCRs).

## 3. Results

In total, 3177 papers matched the choice criteria across 10 document types and three publication types. The 10 document types were article (*n* = 2602), proceedings paper (*n* = 252), review (*n* = 213), conference paper (*n* = 98), editorial material (*n* = 6), correction (*n* = 2), letter (*n* = 1), meeting abstract (*n* = 1), news item (*n* = 1), and book chapter (*n* = 1). The three publication types were journal (*n* = 2929), serials (*n* = 169), and book (*n* = 79). The 3177 papers were published in 690 sources. A total of 8229 authors from 79 countries/regions and 1816 institutions contributed to the research of energy storage ceramics. The vast majority of the articles and reviews were published in English (3140, 98.869%), followed by Chinese (31, 0.976%), Korean (3, 0.098%), Japanese (1, 0.031%), and Spanish (1, 0.031%). An average citation of 24.49 per paper and 77817 total times cited makes the 3177 published papers relevant in the scientific community. A total of 215 funds supported the publication of these papers, and the National Natural Science Foundation of China (*n* = 1225), National Basic Research Program of China (*n* = 223), and Fundamental Research Funds for the Central Universities (*n* = 198) are the top three among them. Including the 3177 papers, there are 105 highly cited papers and five popular papers.

### 3.1. Number of Publications

Figure 1 shows the annual analysis of the published papers. The annual publication numbers grew slowly in the first years analyzed, from 10 (in 2000) to 83 (in 2012). A high growth rate happened in the period 2008–2011, but the yearly production was still less than 100. In the last eight years (from 2013 to 2020), the annual publication number has increased rapidly, rising from 83 papers in 2012 to 680 papers in 2020. The increase in the annual publication number since 2013 could be related to the rise in global energy research. It is also worth noting that there has been a steady increase in annual publications since 2008; the average yearly growth rate was 34.9%. The most productive year was 2020, which increased by 19.3%.

China, the USA, and India are the top three most productive countries. China entered into the field of energy storage ceramics in 2004 and became the leader in 2011. After exceeding the USA, China’s production grew rapidly. The average yearly growth rate was 170% between 2011 and 2020. The USA has a long history of energy storage ceramics research and has been the research center for a long time, until being overtaken by China in 2011. The average percentage growth rate of the USA was 24% in the past 10 years. India took part in the research of energy storage ceramics earlier than China, but not many papers were published until 2018. The yearly production of the USA and India in recent years is approximately 50 papers.

### 3.2. Country-Specific Production and Collaboration

The publications on energy storage ceramics between 2000 and 2020 were derived from 79 countries/regions. As shown in Table 1, the most productive country/region in the energy storage ceramics research field was China, with a publication share of 55.0% (*n* = 1747). The USA ranked second (*n* = 542, 17.1%), followed by India (*n* = 232, 7.3%), Germany (*n* = 177, 5.6%), the UK (*n* = 151, 4.8%), and Japan (*n* = 132, 4.2%). The remaining top 20 most productive countries were mostly located in Asia and Europe. The USA holds the highest average citations of 47.21 per paper, followed by Australia (ACCP = 46.83) and Canada (ACCP = 42.04). Australia (DC = 95.24%), the UK (DC = 90.73%), and Singapore (DC = 89.74%) are the three countries/regions with the highest percentage of papers cited. China (h-index = 91) and the USA (h-index = 74) are ahead of other countries/regions in the field of the h-index.

Figure 2 displays country/region collaborations in energy storage ceramics research. Through the collaboration network, the collaboration relationship with different countries/regions can be more intuitively observed, so as to help find more beneficial collaborators. Each node represented a country/region. The data near the country/region names are the total number of publications from that country/region. The yellow points in the intersections between the countries/regions illustrate collaborative papers with other countries/regions.

The figure shows that China is the leader of energy storage ceramics research in cooperation with other countries/regions, followed by the USA, the UK, and Germany. The most productive countries/regions had more frequent cooperation with other countries/regions. China collaborates with 37 countries/regions with 348 papers, closely linked with the USA, Australia, the UK, Japan, Singapore, Canada, and Russia. The USA has 213 international papers and collaborates with 44 countries/regions, including China, the UK, Japan, and South Korea. India ranks third in the top 20 most productive countries/regions, with 57 international papers. Unlike China and the USA, India’s collaboration with the top 20 most productive countries/regions is not very close. Taiwan is the region with which India collaborates most closely, followed by South Korea and the USA. Among the top 20 most productive countries/regions, Brazil, Thailand, Italy, and Poland have smaller collaboration networks than the other countries/regions. It is worth mentioning that the USA has the largest number of collaborated countries/regions (nCC = 44), and Pakistan has the highest percentage of international collaborations (CC = 77.50%).

### 3.3. Contribution of Leading Institutions

A total of 1816 institutes have participated in energy storage ceramics research. The distribution of institute contributions to publications reiterated the predominance of China in this research field. The top 30 most productive institutes are shown in Table 2. In the table, there are 22 institutions from China; three from the USA; and one from Australia, Germany, India, Singapore, and the UK, respectively. Regarding the top 20 institutions, 18 institutes are located in China, and one in the USA and Australia, respectively. Xi An Jiao Tong Univ ranks first in terms of total publications, followed by Chinese Acad Sci and Tsinghua Univ. Penn State Univ holds the first position for average citations per paper (ACCP = 64.90). Xi An Jiao Tong Univ has the highest h-index value, followed by Chinese Acad Sci and Tsinghua Univ. There is only one institute from Europe, and no institutes from Africa or South America, on this list. It is worth noting that Penn State Univ (ACCP = 64.90) and Univ Wollongong (ACCP = 54.00) are leading in the table of citations per paper, but a large number of researchers from these institutions are from China.

Additionally, we analyzed the collaborations of energy storage ceramics between the top 30 most productive institutions (see Figure 3). Each node represented an institution. The data near the institution names are the total number of publications of the institution. The yellow points in the intersections between the institutions indicate collaborative publications with other institutions in the top 30.

It can be seen that the most productive institutions show more collaboration than other institutions, such as Xi An Jiao Tong Univ, Chinese Acad Sci, Tsinghua Univ, and Wuhan Univ Technol. Among the top 30 most productive institutions, Tsinghua Univ maintains collaboration with more institutions. Including Tsinghua Univ’s 145 papers, 61 papers were collaborated with Univ Wollongong, Univ Sci and Technol Beijing, Chinese Acad Sci, and other top institutions. Chinese Acad Sci has 137 papers collaborated with institutions such as Tsinghua Univ, Tongji Univ, and Univ Elect Sci and Technol, China. Xi An Jiao Tong Univ is the most productive institution, with 106 institution collaborations; Xian Univ Technol and Southwest Univ are the main partners. Other stable collaborative relations include the collaboration between Wuhan Univ Technol and Penn State Univ, and the collaboration between Guilin Univ Elect Technol and Cent S Univ. Among the top 30 most productive institutions, Harbin Inst Technol, Sichuan Unive, Natl Univ Singapore, MIT, Natl Inst Technol, Argonne Natl Lab, and German Aerosp Ctr DL have smaller collaboration networks than other institutions.

### 3.4. Contribution of Leading Authors

The top 20 most productive authors are shown in Table 3; they are mostly from China. Among the 20 authors, there are five from Shaanxi Univ Sci and Technol; three from Xi An Jiao Tong Univ; two from Tongji Univ; and two from Chinese Acad Sci, China. They contributed the largest number of productive authors. Except for one author from Univ Wollongong, the other top 20 most productive authors are all from institutions in China. Here, some close-cooperative teams are represented by a lead author. For example, Wuhan Univ Technol has many productive authors, such as H. Hao, H.X. Liu, M.H. Cao, and Z.H. Yao. They co-authored many papers, and the corresponding author of the most papers is H.X. Liu, so H.X. Liu was the representative of these papers. Additionally, J.W. Zhai represents a research group from Tongji Univ, and X.H. Wang represents a research group from Tsinghua Univ. J.W. Zhai (TP = 86, TC = 2515) is the leader of total productions and citations, followed by H.X. Liu (TP = 73, TC = 2072) and X.L. Dong (TP = 54, TC = 1803). Regarding the average citation per paper, S.J. Zhang ranks first with 68.35, followed by T. Wang (55.19) and H.L. Du (49.86). Z.J. Zhai (25) has the highest h-index value, followed by X.L. Dong (23) and H.X. Liu (21).

### 3.5. Contribution of Leading Research Areas and Journals

Three thousand one hundred and seventy-seven papers related to energy storage ceramics research have been published in 88 SCI research areas, among which the top 20 are listed in Figure 4. Materials science—multidisciplinary (*n* = 1396, 43.96%); physics—applied (*n* = 741, 23.33%); and materials science—ceramics (*n* = 634, 19.96%) are the three research areas with the highest percentage of papers, followed by chemistry—physical (*n* = 616, 19.40%), and energy and fuels (*n* = 446, 14.04%). Research from materials science—multidisciplinary; physics—applied; physics—condensed matter; engineering—electrical electronic; and some other research areas are long term, stable, and focus on the research of energy storage ceramics.

In total, 3177 papers were published in 699 publications, with 407 publications publishing only one paper. In Table 4, the top 30 most productive journals, in terms of the number of publications, categories, and impact factor 2019, are reported. The top 30 journals have published 1662 papers, which represents 52.31% of the papers in this study. *Ceramics International* is ranked first (TP: 285, IF2019: 3.83), *Science-Materials in Electronics* second (TP: 163, IF2019: 2.22), and the *Journal of Alloys and Compounds* third (TP: 158, IF2019: 4.65).

The source growth of the top 30 most productive journals is shown in Figure 5. Except for the top three journals, the *Journal of the European Ceramic Society* (TP: 117, IF2019: 4.495), *Journal of the American Ceramic Society* (TP: 108, IF2019: 3.502), *Journal of Materials Chemistry A* (TP: 84, IF2019: 11.301), *ACS Applied Materials and Interfaces* (TP: 66, IF2019: 8.758), and *Journal of Materials Chemistry C* (TP: 60, IF2019: 7.059) have grown exponentially in recent years. On the contrary, the publication of *Applied Physics Letters*, *International Journal of Hydrogen Energy*, and *Materials Research Express* have declined over time. It is also noteworthy that several journals published papers on energy storage ceramics research during the first 13 years of the 2000s. Since 2013, there have been more publications on energy storage ceramics, indicating that the research area is growing.

### 3.6. An Analysis of Keywords

To study the main direction and trend of energy storage ceramics research, keywords from 3177 papers were analyzed. Due to some papers’ author keywords being missing, here, we used a combination of author keywords and keywords plus to fully reveal this research field. Apart from some of the most commonly used searching keywords, such as “energy storage”, “density”, “ceramics”, “performance”, “energy”, “behavior”, “ferroelectric”, and “dielectric”, the remaining keywords were carefully cleaned. Various expressions of the same subjects, such as “Barium Titanate” and “Batio3”, were merged to ensure that keywords with similar meanings were represented by one unified word. The top 20 cleaned keywords that frequently appeared at the same time are illustrated in Figure 6.

Figure 6 represents a map of energy storage ceramics research. A bubble chart was used to show the development trend of this field in 3D. Using the size of bubble as a third dimension, the chart can be applied to track research frontiers [59]. The number in a bubble represents the frequency of a keyword in that year.

“Microstructure” (*n* = 366) ranks first in terms of occurrence, followed by “thin-films” (*n* = 354) and “phase-transition” (*n* = 301). They are also the top three frequency occurrence keywords. Research processes of the top three keywords are different. Researchers began to focus on ceramic microstructures in 2011. The properties, behavior, characteristics, changes, evolution, modification, and design of microstructures were studied by Z.Y. Shen, A.G. Jain, G. Liu, and other researchers [60,61,62]. Thin films, including ferroelectric thin films and antiferroelectric thin films, are a long-term topic of material research for researchers, such as A. Kumar, Q. Li, and B.H. Ma; relevant theories and methods have been constantly updated in recent years [63,64,65]. Phase-transition was not popular until 2014. With the work of L. Jin, Q. Xu, R. Xu, and other researchers, related work has made great progress in the past seven years [66,67,68].

It is worth noting that some keywords have become frequent in recent years, such as “lead-free ceramics” (since 2017) and “energy storage performance” (since 2016). In 2017, lead-free ceramics became a popular topic; researchers, such as G. Liu, F. Li, and H.B. Yang, published a large number of papers and promoted the research of energy storage ceramics to the lead-free era [69,70,71]. Almost at the same time, the research of energy storage performance became a frequent appearance in keywords; L. Jin, X. Lu, L. Zhang, and other researchers, carried out a series of exploratory works and advanced the topic rapidly [72,73,74].

The following other keywords can also be noted: the research topic of grain size appeared in 2010 and became a frequent keyword in 2014; the effect, engineering, and dependence of grain size were studied by G. Liu, M.S. Alkathy, G. Chen, and other researchers [75,76]; ferroelectric properties is a topic with a long history, and the number of papers has been increasing since 2014 [77,78]; the production of relaxor ferroelectrics research obviously increased in the last three years; researchers, such as G. Liu, F. Li, and Z. Dai, advanced the research of relaxor ferroelectric behavior, polymers, properties, and transition [79,80].

### 3.7. An Analysis of the Most Cited Papers

Among the 3177 papers, there are 105 highly cited papers and five popular papers. Table 5 lists the 20 most cited papers. The earliest paper by Bai et al.—“High-dielectric-constant ceramic-powder polymer composites”—was published by *Applied Physics Letters* in 2000, and described a ceramic-powder polymer composite developed with a high room-temperature dielectric constant relaxor. The most recent paper by Ngo et al.—“Additive manufacturing (3D printing): A review of materials, methods, applications and challenges”—was published by *Composites Part B-Engineering* in 2018, and gave an overview of the main 3D printing methods, materials, and their development in trending applications, and the current state of ceramics materials development was presented. The number of citations ranged from 1941 for Naguib et al.—“25th Anniversary Article: MXenes: A New Family of Two-Dimensional Materials”—to 375 for Yao et al.—“Homogeneous/Inhomogeneous-Structured Dielectrics and their Energy-Storage Performances”. Ngo et al.—“Additive manufacturing (3D printing): A review of materials, methods, applications and challenges”—is ranked first in the field of total citations per year. Five sources, including *Advanced Materials*, *Nature*, *Energy and Environmental Science*, *Materials*, and *Proceedings of the National Academy of Sciences of the United States of America*, published the two most cited papers. The other 10 papers were published in 10 sources, namely, *Composites Part B-Engineering*, *Dalton Transactions*, *Journal of Power Sources*, *Chemical Reviews*, *Nano Energy*, *Applied Physics Letters*, *Advanced Energy Materials*, *Advanced Functional Materials*, and *International Journal of Electrochemical Science*. The USA contributed eleven of them, followed by China (2), Switzerland (1), UK (1), Australia (1), Israel (1), Germany (1), Spain (1), and India (1), which indicated that the USA was the leading country of academic influence in this research field. It is worth noting that many papers are the results of multidisciplinary integration.

### 3.8. An Analysis of Popular Papers

Researchers usually identify the most interesting recent research topics within a research field with popular ESI papers. There were five popular ESI papers in this field, all of which were published in 2019 (Table 6). Three of them are review papers, and two of them are article papers.

L. Yang et al. published their paper in *Progress in Materials Science*, which summarizes the principles and recent developments of perovskite lead-free dielectrics and other types of dielectrics; the new achievements of polymer–ceramic composites in energy-storage applications are also reviewed [97]. A review published in *Chemical Society Reviews*, by H. Luo et al., provides a detailed overview of the latest developments in the design and control of the interface in polymer-based composite dielectrics for energy storage applications, and described efforts to achieve a close control of interfacial properties and geometry, which include the use of liquid crystals, and developing ceramic and carbon-based interfaces with tailored electrical properties [98]. H. Qi et al. published a paper in the *Journal of Materials Chemistry A*, which introduced a novel lead-free polar dielectric ceramic with linear-like polarization responses; BNT-based lead-free AFE ceramic systems may be a potential candidate for application in pulsed power systems [99]. The work of W.G. Ma et al., published in the *Journal of Materials Chemistry C*, used antiferroelectric (AFE) AgNbO_3_ (AN) to partially substitute the relaxor ferroelectric BNT-ST of morphotropic phase boundary (MPB) composition to reduce the remanent polarization [100]. A review in *Energy and Environmental Science*, by Samson et al., discussed the progress and trends in the three main approaches, to realize the technological application of Li_7_La_3_Zr_2_O_12_ (LLZO) as an electrolyte in solid-state Li batteries (SSLBs) [101]. These highly cited papers not only described the latest developments and trends of lead-free dielectrics and other energy storage ceramic materials, but motivate an increasing amount of researchers with multidisciplinary backgrounds to explore potential research areas.

## 4. Discussion

This bibliometric analysis confirmed that energy storage ceramics has become an important component of energy research over the last 20 years. We can divide the research of energy storage ceramics into three stages. Before 2007, it can be called the early accumulation stage; the number of publications and citations was not high, but many topics were discussed. The period from 2007 to 2013 can be called the widespread attention stage. Researchers from various countries and disciplines began to frequently reference energy storage ceramics research papers. With the growth in global energy demand, the period after 2013 can be called the rapid development stage. The number of publications, participating countries, institutions, journals, and researchers has increased significantly, and influence indicators, such as citations and h-index, have also increased significantly. 

China has become the leader of energy storage ceramics research, in terms of the number of publications and h-index, since 2011. Institutions and authors from China are the most productive, with the highest h-index. India, another developing country, is also performing well in this field. However, we can find from other viewpoints, such as average citations per paper and the percentage of international collaborations, that traditional developed countries still have strong research power. Researchers from developed countries, such as the USA, Australia, the UK, and Canada, are still leading the way in energy storage ceramics research. It is also worth noting that major publications are mostly from the USA, UK, Germany, and other developed countries. Emerging countries/regions have a long way to go in the development of science and technology.

We studied the collaborations between institutions and countries/regions. The most productive countries/regions and institutions have a high percentage of collaborations and large number of collaborated institutions. However, no country or institution has become the core of energy storage ceramics and significantly impacted other countries/regions or institutions. Chinese institutions collaborate with many institutions, but their influence is often demonstrated through collaboration with institutions of other countries, such as the USA. The academic influence of the USA is still ranked first, as can be seen from the performance of country/region collaboration and highly cited papers, but the scale of research limits the performance of USA institutions.

The 3177 papers are distributed in 88 research fields, but material science—multidisciplinary; physics—applied; and material science—ceramics hold a large proportion of the total papers. It is also worth noting that papers of material science—ceramics do not significantly grow until after 2013. As an interdisciplinary research area, the subject-integrated level of energy storage ceramics must be improved. As can also be seen from the distribution of publications, *Ceramics International*, and other journals specializing in ceramics, remain the major source of energy storage ceramics papers.

Regarding research topics, lead-free ceramics is the trend of energy storage ceramics; the publication number of related papers has increased rapidly in recent years. Additionally, many frequently used keywords have a relationship with lead-free ceramics. This issue suggests that the increasing requirements of low emissions have inspired an enormous effort towards the development of efficient and clean energy.

## 5. Conclusions

Here, we presented a general overview of energy storage ceramics research, in terms of leading countries/regions, institutes, publications, authors, research fields, highly cited papers, research topics, cooperation, and trends between 2000 and 2020, based on the Web of Science (WOS) databases. 

A total of 3177 publications were identified after retrieval in WOS. From the yearly yield of the field, we can determine the study of energy storage ceramics prosperity from 2013. China definitely had energy storage ceramics research with the most publications and highest h-index. Xi An Jiao Tong University is the most productive institution, with the highest h-index. J.W. Zhai is the most productive and most cited author, with the highest h-index. *Ceramics International Journal of Materials* and another 698 publications published energy storage ceramics research papers in materials science—multidisciplinary and another 87 research areas. “Microstructure”, “thin-films”, and “phase-transition” are the top three topics researchers focused on. The most cited paper has been cited 1941 times, 277.3 times per year. There are 105 highly cited papers and five popular papers; lead-free ceramics is the main research direction of these papers.

This study will help potential energy storage ceramics researchers to quickly understand the global research status of this field. It can also provide relevant researchers with beneficial information on research frontiers, potential collaborators, funding supports, and submission goals of papers. In addition, this work can provide a reference for policymakers to improve energy policies and strengthen energy governance.

## 6. Drawbacks and Future

There were still some drawbacks in our work. The main drawback of our work is that several relevant documents were not covered. The topic search in the Web of Knowledge platform only included the title, abstract, and keywords of a paper; papers with no words matching the search formula may be omitted. These issues will lead to some deviations and affect the results. 

In this work, the research status and development trend of energy storage ceramics in the last 20 years were studied. The development of energy storage ceramics research and bibliometric analysis requires further investigation in many aspects. As an interdisciplinary research field, it is of positive significance for the development of energy storage ceramics research to reveal the status, role and cooperation of materials science, physics, chemistry, energy science, management, and other disciplines. The scientific analysis of the ongoing strength and evolution of popular topics, as well as the prediction of future frontiers, will be helpful for researchers and policymakers.

## Figures and Tables

**Figure 1 materials-14-03605-f001:**
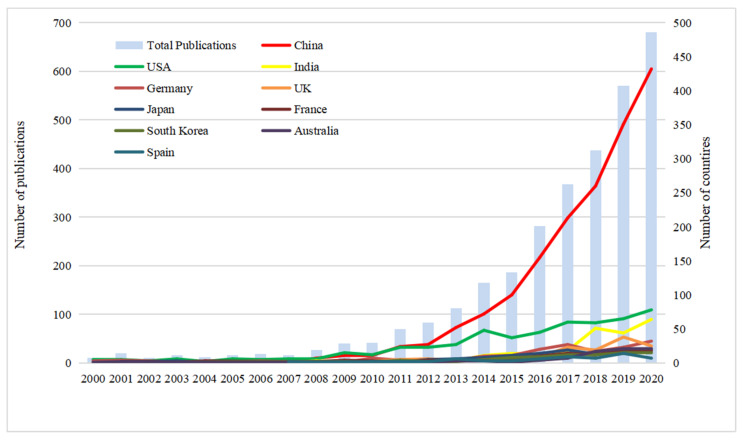
The number of publications of energy storage ceramics research by year.

**Figure 2 materials-14-03605-f002:**
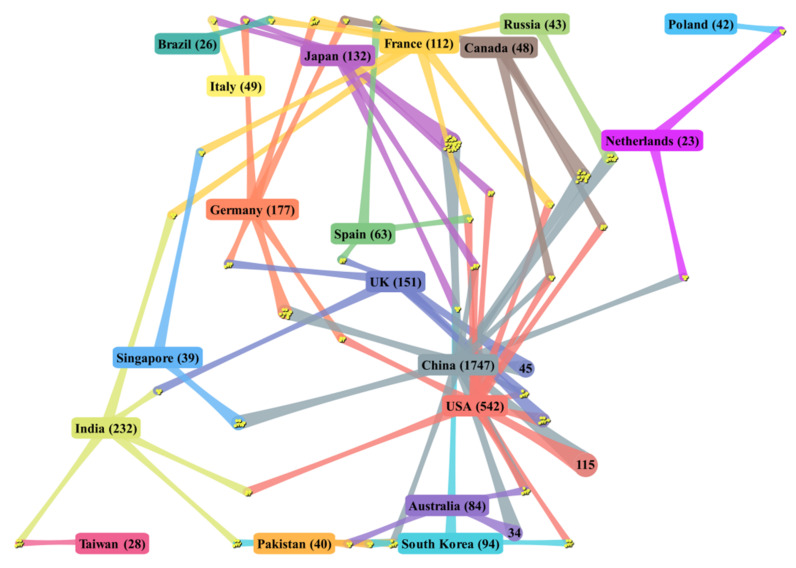
DDA cluster map on collaboration of the top 20 most productive countries/regions.

**Figure 3 materials-14-03605-f003:**
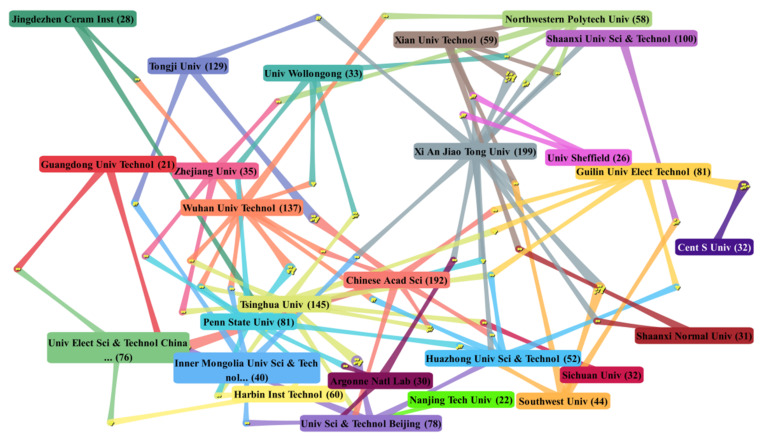
DDA cluster map on collaboration of the top 30 most productive institutions in energy storage ceramics research.

**Figure 4 materials-14-03605-f004:**
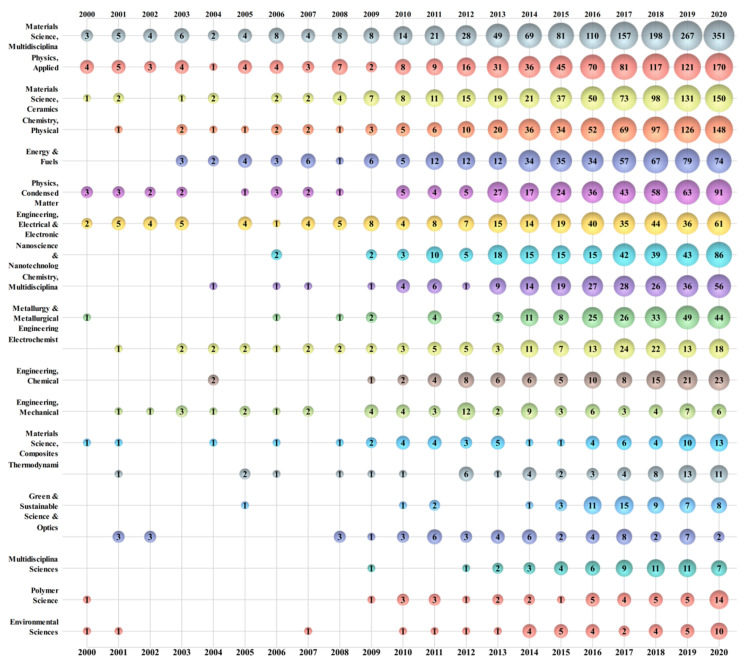
Bubble chart of the top 20 research areas in energy storage ceramics.

**Figure 5 materials-14-03605-f005:**
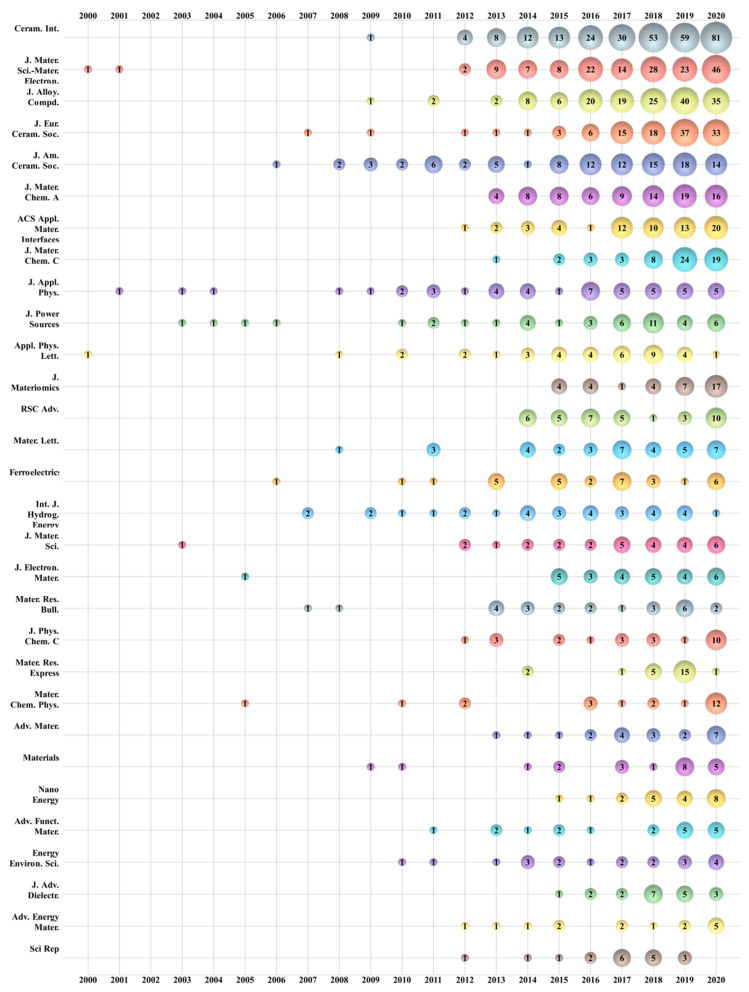
Bubble chart of the top 30 most productive publications by year.

**Figure 6 materials-14-03605-f006:**
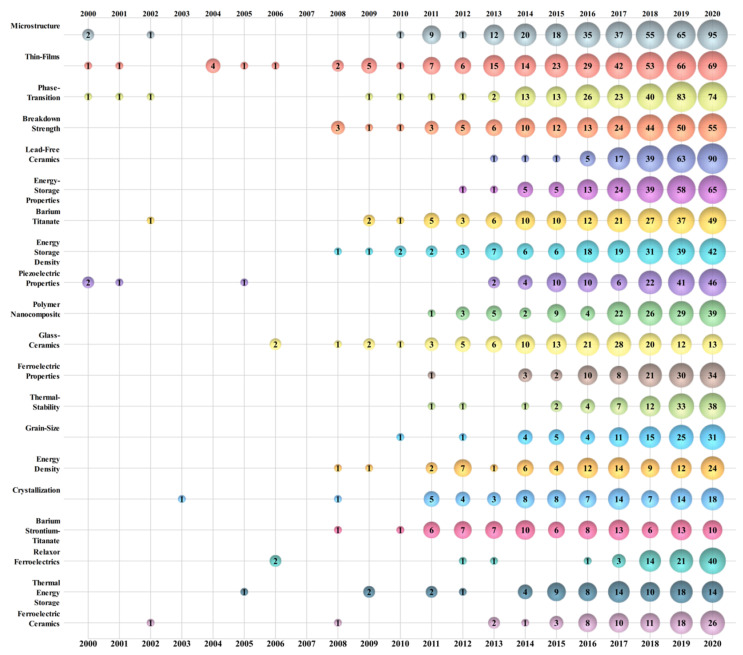
Bubble chart of top 30 keywords of energy storage ceramics research by year.

**Table 1 materials-14-03605-t001:** Contribution and impact of the top 20 most productive countries/regions in energy storage ceramics research.

Rank	Country/Region	TP	TC	ACCP	DC (%)	h-Index	CC (%)	nCC
1	China	1747	38,872	22.25	86.38	91	19.92	37
2	USA	542	25,586	47.21	88.56	74	39.30	44
3	India	232	3145	13.56	75.00	38	24.57	29
4	Germany	177	4581	25.88	83.62	34	45.20	38
5	UK	151	5002	33.13	90.73	36	72.19	31
6	Japan	132	2355	17.84	84.09	28	46.97	27
7	France	112	1540	13.75	83.93	20	58.93	39
8	South Korea	94	1826	19.43	85.11	24	37.23	12
9	Australia	84	3934	46.83	95.24	26	70.24	17
10	Spain	63	1496	23.75	84.13	19	57.14	27
11	Italy	49	833	17.00	85.71	14	55.10	23
12	Canada	48	2018	42.04	85.42	21	72.92	15
13	Russia	43	668	15.53	83.72	11	65.12	16
14	Poland	42	436	10.38	85.71	14	38.10	13
15	Pakistan	40	441	11.03	70.00	9	77.50	16
16	Singapore	39	1424	36.51	89.74	14	61.54	13
17	Taiwan	28	322	11.50	75.00	10	57.14	10
18	Brazil	26	257	9.88	69.23	7	50.00	9
19	Thailand	24	226	9.42	66.67	9	37.50	9
20	Netherlands	23	434	18.87	86.96	10	73.91	18

Note: TP: total paper; TC: total citations; ACCP: average citations per paper; DC%: percentage of papers cited; CC%: percentage of international collaborations; nCC: number of collaborated countries/regions.

**Table 2 materials-14-03605-t002:** The top 30 most productive institutions in energy storage ceramics research during the period 2000–2020.

Institution	TP	TC	ACCP	DC (%)	h-Index	nCI	IC (%)	Country/Region
Xi An Jiao Tong Univ	199	6486	32.59	89.45	41	107	74.87	China
Chinese Acad Sci	192	5848	30.46	86.98	39	121	68.75	China
Tsinghua Univ	145	5451	37.59	88.97	38	86	63.45	China
Wuhan Univ Technol	137	3080	22.48	85.40	29	52	45.99	China
Tongji Univ	129	3419	26.50	93.80	30	47	57.36	China
Shaanxi Univ Sci and Technol	100	2159	21.59	83.00	27	23	26.00	China
Guilin Univ Elect Technol	81	780	9.63	83.95	16	40	48.15	China
Penn State Univ	81	5257	64.90	92.59	31	57	79.01	USA
Univ Sci and Technol Beijing	78	1723	22.09	84.62	19	67	71.79	China
Univ Elect Sci and Technol China	76	953	12.54	92.11	17	96	63.16	China
Harbin Inst Technol	60	913	15.22	83.33	15	38	60.00	China
Xian Univ Technol	59	1065	18.05	86.44	19	39	89.83	China
Northwestern Polytech Univ	58	1054	18.17	87.93	21	29	60.34	China
Huazhong Univ Sci and Technol	52	1417	27.25	88.46	17	45	71.15	China
Southwest Univ	44	343	7.80	81.82	10	26	88.64	China
Inner Mongolia Univ Sci and Technol	40	1170	29.25	90.00	18	19	52.50	China
Zhejiang Univ	35	1014	28.97	85.71	17	26	65.71	China
Univ Wollongong	33	1782	54.00	96.97	16	23	84.85	Australia
Cent S Univ	32	1200	37.50	96.88	14	16	56.25	China
Sichuan Univ	32	670	20.94	81.25	10	23	43.75	China
Shaanxi Normal Univ	31	249	8.03	80.65	9	12	51.61	China
Argonne Natl Lab	30	964	32.13	100.00	13	41	70.00	USA
Jingdezhen Ceram Inst	28	283	10.11	92.86	10	11	46.43	China
Univ Sheffield	26	826	31.77	92.31	15	35	96.15	UK
MIT	24	1001	41.71	100.00	12	32	66.67	USA
German Aerosp Ctr DLR	22	424	19.27	81.82	10	8	27.27	Germany
Nanjing Tech Univ	22	383	17.41	81.82	11	20	95.45	China
Natl Inst Technol	22	235	10.68	81.82	9	38	72.73	India
Natl Univ Singapore	22	807	36.68	86.36	13	22	72.73	Singapore
Guangdong Univ Technol	21	267	12.71	95.24	11	22	66.67	China

Note: TP: total paper; TC: total citations; ACCP: average citations per paper; DC%: the percentage of papers cited; IC%: the percentage of institution collaborations; nCI: number of collaborated institutions.

**Table 3 materials-14-03605-t003:** Contribution of the top 20 authors in energy storage ceramics research.

Rank	Author	TP	TC	nFA	nCA	ACCP	h-Index	Institute
1	Zhai, Jiwei	86	2515	0	76	29.24	25	Tongji Univ (China)
2	Liu, Hanxing	73	2072	0	40	28.38	21	Wuhan Univ Technol (China)
3	Dong, Xianlin	54	1803	0	24	33.39	23	Chinese Acad Sci (China)
4	Jin, Li	54	1648	9	29	30.52	19	Xi An Jiao Tong Univ (China)
5	Pu, Yongping	52	1001	10	37	19.25	20	Shaanxi Univ Sci and Technol (China)
6	Wang, Xiaohui	44	1350	0	40	30.68	17	Tsinghua Univ (China)
7	Wang, Genshui	41	1208	0	26	29.46	19	Chinese Acad Sci (China)
8	Liu, Gang	38	331	13	18	8.71	11	Southwest Univ (China)
9	Hao, Xihong	35	1108	3	29	31.66	16	Inner Mongolia Univ Sci and Technol (China)
10	Fan, Huiqing	33	597	0	19	18.09	16	Northwestern Polytech Univ (China)
11	Yang, Haibo	33	892	11	25	27.03	16	Shaanxi Univ Sci and Technol (China)
12	Zhang, Shujun	31	2119	0	13	68.35	19	Univ Wollongong (Australia)
13	Lin, Ying	30	896	6	8	29.87	17	Shaanxi Univ Sci and Technol (China)
14	Hu, Qingyuan	28	1199	7	2	42.82	18	Xi An Jiao Tong Univ (China)
15	Wang, Tong	26	1435	6	4	55.19	18	Shaanxi Univ Sci and Technol (China)
16	Yang, Tongqing	26	458	0	15	17.62	13	Tongji Univ (China)
17	Zhang, Lei	25	725	9	3	29.00	13	Shaanxi Univ Sci and Technol (China)
18	Cai, Ziming	23	407	12	1	17.70	10	China Univ Min and Technol (China)
19	Du, Hongliang	22	1097	0	15	49.86	16	Xi An Jiao Tong Univ (China)
20	Xu, Jiwen	22	249	1	9	11.32	9	Guilin Univ Elect Technol (China)

Note: TP: total paper; TC: total citations; nFA: number of first author papers; nCA: number of corresponding author papers; ACCP: average citations per paper.

**Table 4 materials-14-03605-t004:** Contribution of the top 30 most productive publications in energy storage ceramics research.

Rank	Publication Name	TP	IF2019	Country/Region	Categories
1	Ceramics International	285	3.83	UK	Materials science, ceramics
2	Journal of Materials Science-Materials in Electronics	163	2.22	Netherlands	Physics, condensed matter physics, applied materials science, multidisciplinary engineering, electrical and electronic
3	Journal of Alloys and Compounds	158	4.65	Switzerland	Chemistry, physical metallurgy and metallurgical engineering materials science, multidisciplinary
4	Journal of the European Ceramic Society	117	4.495	UK	Materials science, ceramics
5	Journal of the American Ceramic Society	108	3.502	USA	Materials science, ceramics
6	Journal of Materials Chemistry A	84	11.301	UK	Energy and fuels chemistry, physical materials science, multidisciplinary
7	ACS Applied Materials and Interfaces	66	8.758	USA	Nanoscience and nanotechnology materials science, multidisciplinary
8	Journal of Materials Chemistry C	60	7.059	UK	Physics, applied materials science, multidisciplinary
9	Journal of Applied Physics	47	2.286	USA	Physics, applied
10	Journal of Power Sources	44	8.247	Netherlands	Energy and fuels chemistry, physical materials science, multidisciplinary electrochemistry
11	Applied Physics Letters	38	3.597	USA	Physics, applied
12	Journal of Materiomics	37	5.797	China Mainland	Chemistry, physical physics, applied materials science, multidisciplinary
13	RSC Advances	37	3.119	UK	Chemistry, multidisciplinary
14	Materials Letters	36	3.204	Netherlands	Physics, applied materials science, multidisciplinary
15	Ferroelectrics	32	0.669	UK	Physics, condensed matter materials science, multidisciplinary
16	International Journal of Hydrogen Energy	32	4.939	UK	Energy and fuels chemistry, physical electrochemistry science
17	Journal of Materials Science	30	3.553	USA	Materials science, multidisciplinary
18	Journal of Electronic Materials	29	1.774	USA	Physics, applied materials science, multidisciplinary engineering, electrical and electronic
19	Materials Research Bulletin	25	4.019	USA	Materials science, multidisciplinary
20	Journal of Physical Chemistry C	24	4.189	USA	Nanoscience and nanotechnology chemistry, physical materials science, multidisciplinary
21	Materials Research Express	24	1.929	UK	Materials science, multidisciplinary
22	Materials Chemistry and Physics	23	3.408	Switzerland	Materials science, multidisciplinary
23	Advanced Materials	22	27.398	Germany (Fed Rep Ger)	Nanoscience and nanotechnology chemistry, physical physics, condensed matter physics, applied materials science, multidisciplinary chemistry, multidisciplinary
24	Materials	22	3.057	Switzerland	Materials science, multidisciplinary
25	Nano Energy	21	16.602	USA	Nanoscience and nanotechnology chemistry, physical physics, applied materials science, multidisciplinary
26	Advanced Functional Materials	20	16.836	Germany (Fed Rep Ger)	Nanoscience and nanotechnology chemistry, physical physics, condensed matter physics, applied materials science, multidisciplinary chemistry, multidisciplinary
27	Energy and Environmental Science	20	30.289	UK	Energy and fuels engineering, chemical environmental sciences chemistry, multidisciplinary
28	Journal of Advanced Dielectrics	20		Singapore	Physics, applied
29	Advanced Energy Materials	19	25.245	Germany (Fed Rep Ger)	Energy and fuels chemistry, physical physics, condensed matter physics, applied materials science, multidisciplinary
30	Scientific Reports	19	3.998	UK	Multidisciplinary sciences

Note: TP: total paper; IF2019: impact factor 2019.

**Table 5 materials-14-03605-t005:** The top 20 most cited publications in energy storage ceramics research field during the period 2000–2020.

Rank	Reference	Title	Source	TC	TC/Year	Institution/Country
1	Naguib M, 2014 [81]	25th Anniversary Article: MXenes: A New Family of Two-Dimensional Materials	Adv. Mater.	1941	277.3	Drexel Univ/USA
2	Manthiram A, 2017	Lithium battery chemistries enabled by solid-state electrolytes	Nat. Rev. Mater.	1114	278.5	Univ Texas Austin/USA
3	Ngo TD, 2018 [82]	Additive manufacturing (3D printing): A review of materials, methods, applications and challenges	Compos. Pt. B-Eng.	924	308	Univ Melbourne/Australia.
4	Stephens FH, 2007 [83]	Ammonia-borane: the hydrogen source par excellence?	Dalton Trans.	788	56.3	LANL/USA
5	Fergus JW, 2010	Ceramic and polymeric solid electrolytes for lithium-ion batteries	J. Power Sources	725	65.9	Auburn Univ/USA
6	Li Q, 2015 [84]	Flexible high-temperature dielectric materials from polymer nanocomposites	Nature	666	111	Penn State Univ/USA.
7	Ligon SC, 2017 [85]	Polymers for 3D Printing and Customized Additive Manufacturing	Chem. Rev.	660	165	Swiss Fed Labs Mat Sci and Technol/Switzerland
8	Sun CW, 2017 [86]	Recent advances in all-solid-state rechargeable lithium batteries	Nano Energy	624	156	Chinese Acad Sci/China
9	Bai Y, 2000 [87]	High-dielectric-constant ceramic-powder polymer composites	Appl. Phys. Lett.	621	29.6	Penn State Univ/USA
10	Lee H, 2014 [88]	A review of recent developments in membrane separators for rechargeable lithium-ion batteries	Energy Environ. Sci.	610	87.1	N Carolina State Univ/USA
11	Crossland EJW, 2013 [89]	Mesoporous TiO_2_ single crystals delivering enhanced mobility and optoelectronic device performance	Nature	601	75.1	Univ Oxford/ UK
12	Barber P, 2009 [90]	Polymer Composite and Nanocomposite Dielectric Materials for Pulse Power Energy Storage	Materials	484	40.3	Univ S Carolina/USA
13	Kraytsberg A, 2012 [91]	Higher, Stronger, Better... A Review of 5 Volt Cathode Materials for Advanced Lithium-Ion Batteries	Adv. Energy Mater.	481	53.4	Technion Israel Inst Technol/Israel
14	Presser V, 2011 [92]	Carbide-Derived Carbon-From Porous Networks to Nanotubes and Graphene	Adv. Funct. Mater.	463	46.3	Drexel Univ/USA
15	Dagdeviren C, 2014 [93]	Conformal piezoelectric energy harvesting and storage from motions of the heart, lung, and diaphragm	Proc. Natl. Acad. Sci. U. S. A.	440	62.9	Univ Illinois/USA
16	Hanemann T, 2010 [94]	Polymer-Nanoparticle Composites: From Synthesis to Modern Applications	Materials	430	39.1	KIT/Germany
17	Jayalakshmi M, 2008 [95]	Simple Capacitors to Supercapacitors—An Overview	Int. J. Electrochem. Sci.	406	31.2	Non Ferrous Mat Technol Dev Ctr NFTDC/India
18	Fu K, 2016 [96]	Flexible, solid-state, ion-conducting membrane with 3D garnet nanofiber networks for lithium batteries	Proc. Natl. Acad. Sci. U. S. A.	395	79	Univ Maryland/USA
19	Hueso KB, 2013	High temperature sodium batteries: status, challenges and future trends	Energy Environ. Sci.	378	47.3	Univ Basque Country/Spain
20	Yao ZH, 2017	Homogeneous/Inhomogeneous-Structured Dielectrics and their Energy-Storage Performances	Adv. Mater.	375	93.8	Wuhan Univ Technol/China

Note: TC: total citations; TC/Year: total citations/year (2020—publication year); Institution/country: institution/country of first corresponding author.

**Table 6 materials-14-03605-t006:** Popular ESI papers in energy storage ceramics research field.

No.	Authors	Article Title	TC	Source	Type	Year
1	L.T. Yang et al.	Perovskite lead-free dielectrics for energy storage applications	196	Prog. Mater. Sci.	Review	2019
2	H. Luo et al.	Interface design for high energy density polymer nanocomposites	124	Chem. Soc. Rev.	Review	2019
3	H. Qi et al.	Linear-like lead-free relaxor antiferroelectric (Bi_0.5_Na_0.5_)TiO_3_-NaNbO_3_ with giant energy-storage density/efficiency and super stability against temperature and frequency	106	J. Mater. Chem. A	Article	2019
4	W.G. Ma et al.	Enhanced energy-storage performance with excellent stability under low electric fields in BNT-ST relaxor ferroelectric ceramics	91	J. Mater. Chem. C	Article	2019
5	A.J. Samson, et al.	A bird’s-eye view of Li-stuffed garnet-type Li_7_La_3_Zr_2_O_12_ ceramic electrolytes for advanced all-solid-state Li batteries	64	Energy Environ. Sci.	Review	2019

Note: TC: total citations; Type: document type.

## Data Availability

All data generated or analyzed during this study are included in this published article.

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
