# Peer review of "Energy Storage Ceramics: A Bibliometric Review of Literature"

_materials, 2021, doi:10.3390/ma14133605_

Round 1

Reviewer 1 Report

The idea of the paper is interesting and the authors perform a splendid job in documenting and researching the existing literature (between 2000 – 2020).

  1. While Fig. 2 – 5 are very interesting are also very hard to read, maybe the authors can enlarge the font a bit.
  2. The authors should consider a statistical analysis to establish trends and future evolution regarding the number of papers and citation. It will be an added value to the paper because other authors will see how hot this topic will be.

Author Response

Question 2.1 While Fig. 2 – 5 are very interesting are also very hard to read, maybe the authors can enlarge the font a bit.

Answer 2.1 We redrew the figures, increased the font size from 12-14 to 16-18. The original figures also will provide for readers to read.

Question 2.2 The authors should consider a statistical analysis to establish trends and future evolution regarding the number of papers and citations. It will be an added value to the paper because other authors will see how hot this topic will be.

Answer 2.2 We thanks the reviewer for this suggestion. We add a paragraph about trends and future evolution regarding the number of papers and citations (see section 4, line 393).

Reviewer 2 Report

This paper shows a good review of an overview of energy storage ceramics research from aspects of types of publications. There are some issues that need to address:

- Fig. 1 needs to add 10 top countries in this image.

-Quality figs. 2-6 are poor, details hard to read.

- There are some grammatical errors, please carefully check the whole manuscript.

- section of drawbacks and future could be increased quality of the manuscript.

-Using a source (WOS) to compile a review work with this level of work is not correct, the authors should point out in the text of the article that the use of one source cannot indicate the validity of the results. More resources (Scopus, Google and, etc.) could help improve this article.

Author Response

Question 3.1 Fig. 1 needs to add 10 top countries in this image.

Answer 3.1 We thanks the reviewer for this suggestion. We redrew the figure and add 10 top countries (see line 151).

Question 3.2 Quality figs. 2-6 are poor, details hard to read.

Answer 3.2 We redrew the figures, increased the font size from 12-14 to 16-18. The original figures also will provide for readers to read.

Question 3.3 There are some grammatical errors, please carefully check the whole manuscript.

Answer 3.3 There are indeed some grammatical errors in the manuscript. We carefully reviewed the paper and used MDPI editing services for extensive English revisions.

Question 3.4 section of drawbacks and future could be increased quality of the manuscript.

Answer 3.4 We thanks the reviewer for this suggestion. We added a section of drawbacks and future after conclusion (see section 6, line 454).

Question 3.5 Using a source (WOS) to compile a review work with this level of work is not correct, the authors should point out in the text of the article that the use of one source cannot indicate the validity of the results. More resources (Scopus, Google and, etc.) could help improve this article.

Answer 3.5 We thanks the reviewer for this suggestion. In our work, we included many relevant databases, including SCI, ESCI, and CPCI-S to ensure adequate recall rate. We have retrieved in Scopus, Google Scholar, Baidu Scholar with the search formula of "energy storage ceramic*" or "lead-free ceramic*" or "dielectric ceramic*". The result showed that the number of documents retrieved in Scopus, Google Scholar, and Baidu Scholar is not necessarily much more than WOS. Many documents not included in WOS are commercial journal papers or papers with low citations. We have comprehensively compared Scopus and WOS. Scopus has collected 39000 journals and conferences. In WOS, 13000 journals are included in SCI, 9000 journals are included in ESCI, and a large number of conference documents are in CPCI-S. As a result, there is little difference in the number of documents between Scopus and WOS, their material research journal collection has 95% repeatability. The advantage of Scopus is that it contains more languages and more conference papers, but its citation analysis function is weaker than WOS in terms of bibliometric research, so our manuscript chose WOS instead of Scopus.

Reviewer 3 Report

The manuscript is very useful and well organized. I don't have any significant remarks that would be improve the quality of the paper. I recommend acceptance.

Author Response

Thanks very much to the reviewer.

Reviewer 4 Report

The paper entitled "Energy storage ceramics: a bibliometric review of literature" provides a bibliometric analysis in order to evaluate publications in the energy storage ceramics field between 2000 and 2020 based on the Web of Science records. The research offers a detailed overview of energy storage ceramics research from aspects of types of documents, citations, h-indices, publish years, publications, institutions, countries/regions, research areas, highly cited papers, and keywords. The paper is well-written and yields interesting results, even though some figures (e.g. Figure 1 or Figure 4) can be re-done in order not to look blurry and difficult to follow. Below are some of my comments and suggestions for the authors:

  1. In the abstract, please correct the "web of science (WOS) databases".
  2. Why was WoS chosen over, for example, Scopus? Scopus is a product of Elsevier and I expect that many journals focusing on material will be indexed there as well.
  3. How can be the result that "China is the leader of energy storage ceramics research" interpreted? Perhaps, this is because Chinese scholars are trying to outperform American and EU scholars in terms of published papers? What about the quality of these papers and the journals they are published at? Is it a "quantity over quality" issue?
  4. Figure 4 is difficult to read and follow.
  5. Conclusions need to be extended to include the limitations of the study and pathways for further research.
  6. The paper needs a thorough English proofreading: there are many minor flaws and some punctuation issues.

Author Response

Question 5.1 In the abstract, please correct the "web of science (WOS) databases".

Answer 5.1 We thanks the reviewer for this suggestion. We have revised the manuscript "based on the Web of Science (WOS)" (see line 13).

Question 5.2 Why was WoS chosen over, for example, Scopus? Scopus is a product of Elsevier and I expect that many journals focusing on material will be indexed there as well.

Answer 5.2 We thanks the reviewer for this suggestion. In our work, we included many relevant databases, including SCI, ESCI, and CPCI-S to ensure an adequate recall rate. We have retrieved in Scopus, Google Scholar, Baidu Scholar with the search formula of "energy storage ceramic*" or "lead-free ceramic*" or "dielectric ceramic*". The result showed that the number of documents retrieved in Scopus, Google Scholar, and Baidu Scholar is not necessarily much more than WOS. Many documents not included in WOS are commercial journal papers or papers with low citations. We have compared Scopus and WOS in a comprehensive way. Scopus has collected 39000 journals and conferences. In WOS, 13000 journals are included in SCI, 9000 journals are included in ESCI, and a large number of conference documents are in CPCI-S. As a result, there is little difference in the number of documents between Scopus and WOS, their material research journal collection has 95% repeatability. The advantage of Scopus is that it contains more languages and more conference papers, but its citation analysis function is weaker than WOS in terms of bibliometric research, so our manuscript chose WOS instead of Scopus.

Question 5.3 How can be the result that "China is the leader of energy storage ceramics research" interpreted? Perhaps, this is because Chinese scholars are trying to outperform American and EU scholars in terms of published papers? What about the quality of these papers and the journals they are published at? Is it a "quantity over quality" issue?

Answer 5.3 We thanks the reviewer for this suggestion. China does not have advantages in all indicators of energy storage ceramics research, for example, it lags behind the United States in terms of average citations per paper. We conclude that "China is the leader of energy storage ceramics research" were from the aspects of number of publications, citations, institutions and h-index. In line 404 of this manuscript, we revised it as "China has become the leading of energy storage ceramics research in term of number of publication and h-index since 2011".

Question 5.4 Figure 4 is difficult to read and follow.

Answer 5.4 We redrew the figure, increased the font size to 18 (see line 264). The original figure also will provide for readers to read.

Question 5.5 Conclusions need to be extended to include the limitations of the study and pathways for further research.

Answer 5.5 We thanks the reviewer for this suggestion. We added a section of drawbacks and future after the section of conclusion (see section 6k, line 454).

Question 5.6 The paper needs a thorough English proofreading: there are many minor flaws and some punctuation issues. Confident in your favourable opinion, I send you my most cordial greetings.

Answer 5.6 We carefully reviewed the paper and used MDPI editing services for extensive English revisions.

Round 2

Reviewer 2 Report

The paper has been improved and corresponding modifications have been conducted. I think the current version can be considered for publication.